# New-Onset Atrial Fibrillation and Early Mortality Rate in COVID-19 Patients: Association with IL-6 Serum Levels and Respiratory Distress

**DOI:** 10.3390/medicina58040530

**Published:** 2022-04-11

**Authors:** Gianluca Bagnato, Egidio Imbalzano, Caterina Oriana Aragona, Carmelo Ioppolo, Pierpaolo Di Micco, Daniela La Rosa, Francesco Costa, Antonio Micari, Simona Tomeo, Natalia Zirilli, Angela Sciacqua, Tommaso D’Angelo, Irene Cacciola, Alessandra Bitto, Natasha Irrera, Vincenzo Russo, William Neal Roberts, Sebastiano Gangemi, Antonio Giovanni Versace

**Affiliations:** 1Department of Clinical and Experimental Medicine, University of Messina, 98125 Messina, Italy; gianbagnato@gmail.com (G.B.); oriana.aragona@hotmail.it (C.O.A.); carmelo.ioppolo@polime.it (C.I.); daniela.larosa@polime.it (D.L.R.); simona.tomeo@polime.it (S.T.); natalia.zirilli@polime.it (N.Z.); irene.cacciola@unime.it (I.C.); abitto@unime.it (A.B.); nirrera@unime.it (N.I.); gangemis@unime.it (S.G.); agversace@unime.it (A.G.V.); 2Department of Medicine, Buonconsiglio Fatebenefratelli Hospital, 80122 Naples, Italy; pdimicco@libero.it; 3BIOMORF Department, University of Messina, 98125 Messina, Italy; dottfrancescocosta@gmail.com (F.C.); antonio.micari@unime.it (A.M.); tommasodang@gmail.com (T.D.); 4Department of Medical and Surgical Sciences, University Magna Græcia of Catanzaro, 88100 Catanzaro, Italy; sciacqua@unicz.it; 5Department of Medical Translational Sciences, Division of Cardiology, Monaldi Hospital, University of Campania “Luigi Vanvitelli”, 80131 Naples, Italy; vincenzo.russo@unicampania.it; 6Department of Medicine, University of Kentucky, Lexington, KY 40506, USA; neal.roberts@uky.edu

**Keywords:** COVID-19, atrial fibrillation, mortality risk, interleukin 6

## Abstract

*Background and objectives*: COVID-19 is associated with an aberrant inflammatory response that may trigger new-onset cardiac arrhythmias. The aim of this study was to assess the mortality risk in hospitalized COVID-19 patients according to IL-6 serum levels and new-onset atrial fibrillation (AF) according to PaO^2^/FiO^2^ stratification. *Materials and Methods*: 175 COVID-19 patients (25 new-onset AF, 22 other types of AF and 128 no-AF) were included in this single-center, retrospective study; clinical and demographic data, vital signs, electrocardiograms and laboratory results were collected and analyzed. The primary outcome of the study was to evaluate the mortality rate in new-onset AF patients according to IL-6 serum levels and PaO^2^/FiO^2^ stratification. *Results:* The incidence of new-onset AF in the study population was 14.2%. Compared to the no-AF group, new-onset AF patients were older with a positive history of chronic kidney disease and heart failure, had higher IL-6, creatinine and urea serum levels whereas their platelet count was reduced. After PaO^2^/FiO^2^ stratification, 5-days mortality rate was higher in new-onset AF patients compared to patients with other types of AF and no-AF patients, and mortality risk increases 5.3 fold compared to no-AF (*p* = 0.0014) and 4.8 fold compared to other forms of AF (*p* = 0.03). *Conclusions:* New-onset AF is common in COVID-19 patients and is associated with increased IL-6 serum levels and early mortality. Further studies are needed to support the use of IL-6 as an early molecular target for COVID-19 patients to reduce their high rate of mortality.

## 1. Introduction

Coronavirus disease 2019 (COVID-19), a new infectious disease induced by severe acute respiratory syndrome coronavirus 2 (SARS-CoV-2), has reached the pandemic status challenging the health systems worldwide [1]. The severity of clinical manifestations in patients with COVID-19 are mainly related to the degree of respiratory system failure ranging from mild cases to life-threatening acute respiratory distress syndrome (ARDS) [2].

In addition, increasing evidence support the application of predictive scores based on several systemic biomarkers, thus suggesting that COVID-19 is characterized by multi-organ complications [3,4,5,6]. Among these, it has been reported that about 12% of patients with COVID-19 develop cardiac injury, such as ischemic and non-ischemic myocardial damage, pericardial disease and arrhythmias [7,8,9].

Several studies confirm an increased risk of developing new onset atrial fibrillation (AF) in patients affected by COVID-19, ranging between 10 and 18%, with a cumulative incidence of all atrial arrhythmias around 27.5% [10,11,12,13]. Indeed, a machine learning approach study confirms that COVID-19 is the strongest independent factor associated with incident AF, compared to the traditional cardiovascular co-morbidities such as congestive heart failure and coronary artery disease [14]. AF may develop in response to several conditions in COVID-19, such as direct myocardial injury, extra-cardiac clinical events in the early phase of the disease, and the inflammatory state *per se*.

Apart from typical presenting clinical scenario, represented by fever, shortness of breath and cough, AF can be the first presenting sign prior to evident respiratory distress, thus reducing the suspicion for this viral infection [15,16]. Indeed, a study involving 137 COVID-19 patients confirmed that 7.3% of these patients experienced palpitations as one of their presenting symptoms [17], probably as a consequence of COVID-19 since not all COVID-19 patients have history of previous cardiac arrhythmia; in fact, another study demonstrated that 4% to 6.6% of COVID-19 patients without prior history of AF exhibited a new diagnosis of atrial arrhythmia during hospitalization [18]. Available studies suggest that the AF incidence may vary with respect to the study population. Importantly, it also remains to clarify whether the inflammatory response activated during COVID-19 is uniquely responsible for AF, or whether it reflects part of a nonspecific immune response of the disease severity.

The aim of the present study was to analyze the impact of IL-6 and new-onset AF on mortality rate in hospitalized COVID-19 patients stratified for PaO^2^/FiO^2^.

## 2. Materials and Methods

### 2.1. Study Population

Adult patients (≥18 years of age) with laboratory-confirmed COVID-19 infection, admitted to the COVID-19 division of the University of Messina between January 2021 and September 2021, were included in this single-center, retrospective study.

In particular, patients who received a COVID-19 diagnosis via polymerase chain reaction (PCR) tests and evidence of pneumonia assessed by computerized tomography (CT) within 24 h of hospital admission were considered eligible for the study (*n* = 190).

Moreover, patients who were either admitted with non-COVID-19 symptoms and had incidental asymptomatic diagnosis (e.g., hip fracture without clinical features of COVID-19, *n* = 5), those admitted for isolation/control reasons (*n* = 3), and also those with COVID-19 who were discharged from either ambulatory care or the emergency room without admission (*n* = 3) were also excluded. Additionally, COVID-19 patients with AF and moderate/severe mitral stenosis (*n* = 2) and with mechanical prosthetic heart valves (*n* = 2) were excluded.

Based on these inclusion and exclusion criteria, a total of 175 COVID-19 patients were enrolled in this study. All COVID-19 patients underwent serial 12-lead electrocardiogram (ECG) at admission and were classified, according to ESC guidelines [19], as either new-onset AF (AF not diagnosed before, irrespective of its duration or the presence of AF-related symptoms; *n* = 25), other AF (paroxysmal, persistent, permanent or long-standing permanent; *n* = 22) or no AF (*n* = 128) based on ECG at admission and history of AF. Patients were further stratified according to PaO^2^/FiO^2^ to identify patients with severe respiratory failure at admission [20]. All patient data were de-identified before analysis and the study AF (365 n. 79-20/11032021).

### 2.2. Data Collection

Data were collected from the electronic health records including baseline demographics, vital signs, clinical presentation, laboratory measurements and outcome of interest at admission. HAS-BLED and CHAD2VASC2 score were recorded for all AF patients.

AF diagnosis was confirmed and verified by two board certified cardiac electrophysiologists who reviewed all ECGs. Other available ECGs were independently reviewed by a cardiologist or electrophysiologist and chart documentation was assessed for atrial fibrillation.

At admission, patients with new onset AF were treated either with heparin or oral anticoagulant according to clinical and laboratory data, while patients with other forms of AF were treated according to the therapeutic regimen already in use. Heparin in a prophylactic dosage was prescribed for COVID-19 patients without any form of AF. Therapeutic regimen related to steroid use (dexamethasone 6 mg/day or equivalent) and oxygen supplementation (conventional oxygen therapy, high-flow nasal oxygen, continuous positive airway pressure and non-invasive mechanical ventilation) have been recorded.

### 2.3. Statistical Analysis

Statistical analyses were performed using SPSS version 21.0 (SPSS Inc., Chicago, IL, USA) software. The distribution of quantitative variables was checked with the Shapiro Wilk test. Descriptive data was given as mean ± standard deviation or median (interquartile range, IQR) according to the distribution normality. The comparison of the distributions of the variables was performed with ANOVA test with post-hoc Fisher’s least significant difference (LSD) or by Kruskal-Wallis with Dunn’s post hoc tests.

Independent T-test was used for the comparison of normally distributed quantitative variables or the Mann-Whitney U test for the comparison of non-normally distributed quantitative variables. Mann-Whitney U test or Fisher’s exact test were used to compare differences between groups. Other analyses included simple and relative frequencies, association tests among categorical variables by χ2 test (*p* < 0.05) and hazard ratio with 95% confidence intervals (95% CI).

In the survival analysis, the outcome was tested related to the event (discharge/death) restricted at 28 days. The outcome (discharge/death) was also tested censoring events after the 5th day of hospital stay to assess early mortality. Thus, it was possible to calculate the probability of COVID-19 death during the entire hospital stay and also during the first 5 days of admission. The Kaplan–Meier survival function was used with the log-rank test to plot survival curves and to determine differences in survival rates, considered different when *p* < 0.05. Cox proportional hazards model [16] was used to investigate the association between the survival time of patients and the set of predictor variables (creatinine and IL-6 serum levels as continuous variables, new-onset AF and PaO^2^/FiO^2^ < 300 as categorical variables). The outcome was time to death, defined as the number of days between admission and death, calculated without censoring and by censoring after 5 days to detect also very early in-hospital mortality. According to the number of events (*n* = 45), we included four variables (PaO^2^/FiO^2^, IL-6, creatinine and new-onset AF) for the identification of independent predictors of the outcome. The Hazard Ratio (HR) with 95% confidence interval (CI) was measured to identify risk factors (the effect of a variable) on the outcome of interest.

## 3. Results

A total of 175 consecutive patients affected by COVID-19 were enrolled: 14.2% with new-onset AF (*n* = 25),12.6% with other forms of AF (*n* = 22) and 73.2% with no AF (*n* = 128). The main demographic and clinical characteristics of the study population at admission are shown in Table 1, while laboratory parameters are reported in Table 2.

### 3.1. Clinical Profile and Mortality Rate of the Study Population

In our study, 28-days mortality rate was 52% for new-onset AF (*n* = 13), 22% for other forms of AF (*n* = 5) and 18% for no AF patients (*n* = 23). Among acute events at admission, ischemic stroke was significantly more frequent in new-onset AF compared to the other groups (new onset AF: 16%, other AF: 4.5% and no AF: 0.7%; *p* = 0.026). On the other hand, no significant differences were observed in myocardial infarction incidence among the study population (new onset AF: 8%, other AF: 4.5% and no AF: 3.1%; *p* = 0.315).

Baseline embolic risk, as assessed by CHADS2-VASc2, and baseline bleeding risk (HAS-BLED) did not show significant differences between new-onset AF and other forms of AF (Table 1).

Clinical profile of the study population was evaluated as initial analysis. Both patients with new-onset AF and other forms of AF were older than no AF group (*p* < 0.02 and *p* < 0.01 respectively), while no significant differences were observed between new-onset AF and other forms of AF. History of heart failure was more frequently associated with the other forms of AF when compared to the no AF group (*p* < 0.005), while history of chronic kidney disease was more frequent both in new-onset AF and in other forms AF compared to the no AF group (*p* < 0.005 for both). New-onset AF patients had significantly higher respiratory rate at admission compared to both groups, while fever was a common symptom in the no AF group. No differences were detected in PaO^2^/FiO^2^ ratio, steroid use and oxygen therapy between all groups (Table 1).

### 3.2. IL-6 Is Increased in New Onset AF and Is Associated with Specific Laboratory Parameters

Analysis of the laboratory parameters revealed that new-onset AF patients had significantly higher IL-6, creatinine and urea serum levels compared to the other groups. In addition, platelet count was significantly lower in the new-onset AF group compared to no AF group (Figure 1A–D). No differences were observed in the other laboratory results. In addition, in the new-onset AF group platelet count inversely correlated with IL-6 levels whereas creatinine and urea were directly associated with IL-6 levels (Figure 2A–C). 

### 3.3. PaO^2^/FiO^2^ Stratification and Mortality

Considering the natural history of COVID-19 and the occurrence of respiratory failure, we stratified the study population according to PaO^2^/FiO^2^ at admission. After applying the cut-off of 300 for PaO^2^/FiO^2^ to identify mild to moderate ARDS, COVID-19 patients with a PaO^2^/FiO^2^ > 300 were excluded from further analysis.

Thus, 25 patients belonging to the no AF group, 4 patients from the other AF group and 6 patients from the new-onset AF group were excluded. After PaO^2^/FiO^2^ stratification, the mortality rate in patients with mild to moderate ARDS (PaO^2^/FiO^2^ between 200 and 300) was 57% (*n* = 11) for new-onset AF, 22% (*n* = 4) for the other AF group and 20% (*n* = 21) for the no AF group.

A significant low survival probability was detected according to Kaplan Meier curves for new-onset AF patients, mostly noted in the first 5 days of admission (Figure 3A–C). Indeed, the 5-days mortality risk after excluding patients with PaO^2^/FiO^2^ > 300, is 5.3 fold higher in new onset AF patients than no AF group (95% CI: 1.91–15.02; *p* = 0.0014) and similarly 4.8 fold higher compared to patients with other forms of AF (95% CI: 1.14–20.2, *p* = 0.03).

Hazard ratio showed that 5-days survival, calculated in patients with PaO^2^/FiO^2^ < 300 at admission, was reduced by 6.09 fold (95% CI 2.69–13.8; *p* = 0.0001) in new-onset AF patients compared to no AF patients and by 4.26 fold (95% CI 1.06–17.11; *p* = 0.04) compared to other AF patients. As previously analyzed for the entire study population, IL-6 levels were also compared between the groups according to PaO^2^/FiO^2^ stratification and they were still significantly higher in new-onset AF patients compared to no-AF group (Figure 3C).

### 3.4. New Onset AF, IL-6, Creatinine and PaO^2^/FiO^2^ < 300 Are Independent Predictors of Mortality in COVID-19 Patients

After stratifying patients, we re-analyzed the mortality risk in the entire study population to identify independent predictors of mortality. Without PaO2/FiO2 stratification, new-onset AF patients showed a 4.94 folds increase in mortality risk compared to no AF patients (95% CI: 2–12.22, *p* = 0.0005) and 3.68 folds increase compared to other forms of AF (95% CI: 1.03–13.10; *p* = 0.04).

A significant difference was observed in survival time obtained by Kaplan-Meier curves between the groups (Figure 4A, *p* = 0.0001): new-onset AF showed a lower probability to survive compared to the other groups. This result is further supported by the sub-analysis at 5 days that showed the relevant reduction of survival probability for new-onset AF patients (Figure 4B). Indeed, hazard ratio demonstrated that 5-days survival probability was reduced by 4.65 fold in new-onset AF patients (95% CI 2.21–9.76; *p* = 0.0001) compared to no AF patients and by 4.4 fold (95% CI 1.07–17.9; *p* = 0.03) compared to other AF patients.

Next, a Cox regression model was fitted considering the entire study population to assess the prognostic value of IL-6 and creatinine serum levels, new-onset AF and PaO^2^/FiO^2^ < 300. Cox regression results showed that IL-6 and creatinine serum levels, new onset AF and PaO^2^/FiO^2^ < 300 are independent predictors of both 5-days and 28-days mortality (Figure 4B–D).

## 4. Discussion

The new infectious disease induced by severe acute respiratory syndrome coronavirus 2 (SARS-CoV-2) has infected, as of April 2022, almost 500 million people causing the death of over 6 million [21]. COVID-19 course is hampered by the occurrence of cardiac arrhythmias and AF represents the most frequent complication, strikingly increasing the risk of mortality from 22 to 56% as well as raising the question about long term consequences in survivors [9,22,23,24,25,26,27]. Similarly, the incidence of new-onset AF in non-cardiac critically ill patients without COVID19 is about 5–15%, leading to poor outcomes compared to patients without AF [28]. Several studies have demonstrated that mortality is an early event in COVID-19 patients with new-onset AF [18,29,30,31] as previously reported in non-COVID-19 patients with pneumonia, sepsis and other viral diseases [32,33,34,35,36]. Our results further extend previous evidence that among COVID-19 patients with AF, those having new-onset AF have a lower chance of survival with a very early mortality, occurring during the first 5 days of hospitalization, compared to both COVID-19 patients with other types of AF and those without AF.

The occurrence of very early events might have different possible explanations, such as the relationship between viral infections and an early aberrant inflammatory response; in this context, also AF onset has been widely described, although not completely defined [37]. Cytokine storm activated during COVID-19 as well as oxidative stress and atrial remodeling processes are among the putative events that may induce AF onset [38]. In addition, the inflammatory cascade might contribute to the development of an arrhythmogenic milieu by inducing complex alterations in atrial myocardiocytes [39].The prominent role of IL-6 in atrial fibrillation patients without COVID-19 has been recently linked to early neurological deterioration, defined as a ∆ ≥ 4 points in the National Institute of Health Stroke Scale (NIHSS) between admission and after 24 h. Indeed, in the cohort of patients included in this study, a higher proportion of COVID-19 patients with new-onset AF had an acute ischemic stroke at admission, as well as high levels of IL-6 compared to the other groups [40]. However, the specific IL-6 trans-signaling related to neurologic events in new-onset AF patients needs to be further identified [41,42].

The association between a pro-inflammatory status and the onset of AF in viral diseases has been evaluated by a recent study in which the authors noted that new onset AF in COVID-19 patients might be attributed to the increased release of pro-inflammatory cytokines, mostly IL-6, as observed in the present study. In addition, the observation that new onset AF occurs in COVID-19 at a frequency similar to that observed in influenza could rule out a specific viral mechanism affecting atrial rhythm [18]. Conversely, some investigators have hypothesized that new onset AF might induce an increase in IL-6 serum levels leading to pulmonary vascular dysfunction and cardiac immune response [43] in subjects with respiratory distress and hypoxemia due to COVID-19 [18], but confirmatory studies are lacking. To our knowledge, this is the first study aiming to assess IL-6 serum levels in COVID-19 patients with and without AF and stratified according to PaO^2^/FiO^2^. Despite no changes were detected in survival rates after stratification, IL-6 levels increased in every group, without differences between new-onset AF and other forms of AF.

Accumulating evidence suggest that COVID-19 predisposes patients to embolic disorders, but a direct association between SARS-CoV-2 and platelet dysregulation has not been fully explained. Our results show that COVID-19 patients with new-onset AF have reduced platelet count, as previously reported [18]. This condition is probably due to a direct viral effect on platelets which may induce their activation, thus potentiating their prothrombotic and inflammatory function via Spike/ACE2 interactions [44,45]. Moreover, IL-6 might directly contribute to the proatherogenic process by stimulating vascular smooth muscle proliferation, endothelial cell activation and platelet activation [46,47].

The complex interaction between immune activation and vascular damage in patients with AF and COVID-19 is further supported by the reported increased prevalence of chronic kidney disease [48,49,50], as observed in our study. In fact, Russo et al. showed an association between chronic kidney disease and incident AF in patients admitted in the emergency units for COVID-19 disease but this finding was not confirmed [27]. Our study confirms that, at admission, patients with new-onset AF had increased creatinine serum levels, suggesting that the onset of this arrhythmia might induce hemodynamic instability, leading to reduced renal perfusion.

Nonetheless, our results suggest that new-onset AF, IL-6 and creatinine levels and PaO^2^/FiO^2^ < 300 represent independent predictors of mortality. These findings suggest that the increased in-hospital mortality is dependent on the severity of the critical disease and worsens in the presence of new-onset AF, kidney dysfunction and increasing IL-6 serum levels.

However, the present study has several limitations. First, the results were collected from a single center without any validation in a second population. Second, it is impossible to assess whether new-onset AF is related to a direct COVID-19 cardiac involvement or due to critical illness itself, due both to the limits of the retrospective study design and the number of AF patients in each group. AF incidence in our cohort is similar to that reported in other studies; however, the present study highlights that new-onset AF is related to IL-6 serum levels in COVID-19 patients even when stratified according to ARDS status.

## 5. Conclusions

In conclusion, new-onset AF is a relevant event that needs to be taken into serious consideration during the assessment of COVID-19. Furthermore, due to the occurrence of very early mortality in this group of patients, and the increase of IL-6, further studies are needed to determine if new-onset AF alone could be considered a relevant marker for better prognostic stratification that might contribute to very early mortality rates.

## Figures and Tables

**Figure 1 medicina-58-00530-f001:**
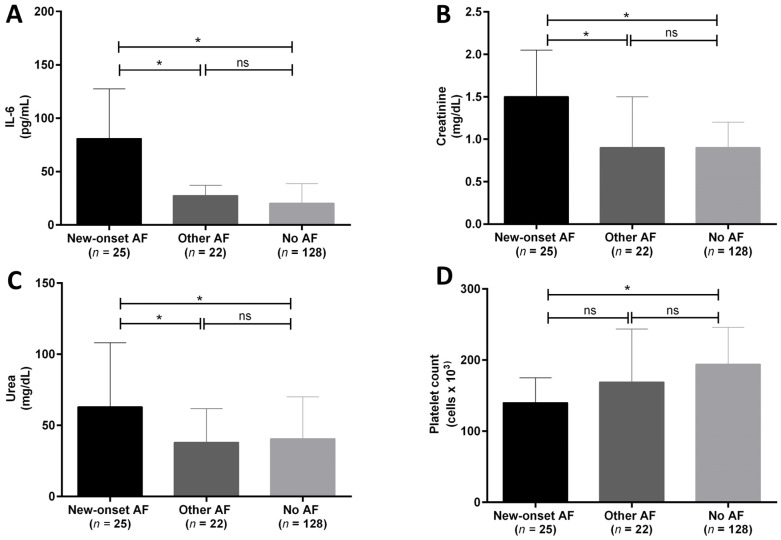
(**A**–**D**). Between-groups analysis of IL-6 (**A**), creatinine (**B**), urea (**C**) levels, and platelet count (**D**), after ANOVA results (see Table 2) in hospitalized COVID-19 patients with new-onset atrial fibrillation (new-onset AF, *n* = 25, black column), with other forms of AF (other AF, *n* = 22, dark grey column) and without AF (no AF, *n* = 128, light grey column). Data are expressed as median ± interquartile range. * = *p* < 0.5; ns = not significant.

**Figure 2 medicina-58-00530-f002:**
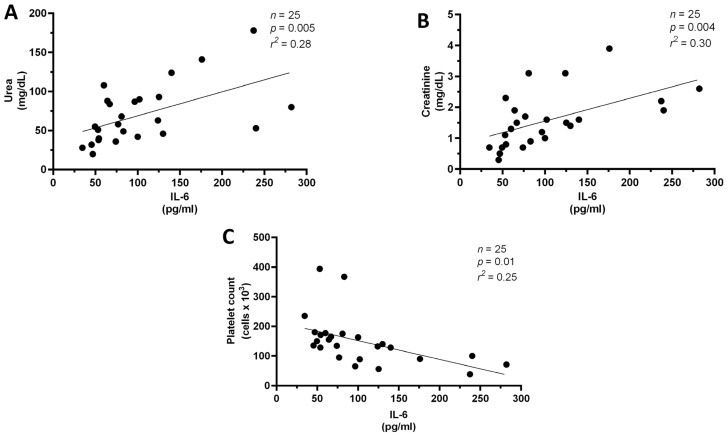
(**A**–**C**). Associations between IL-6 and kidney function (creatinine and urea, **A**,**B**) and platelet count (**C**) in hospitalized COVID-19 patients with new-onset atrial fibrillation (new-onset AF, *n* = 25).

**Figure 3 medicina-58-00530-f003:**
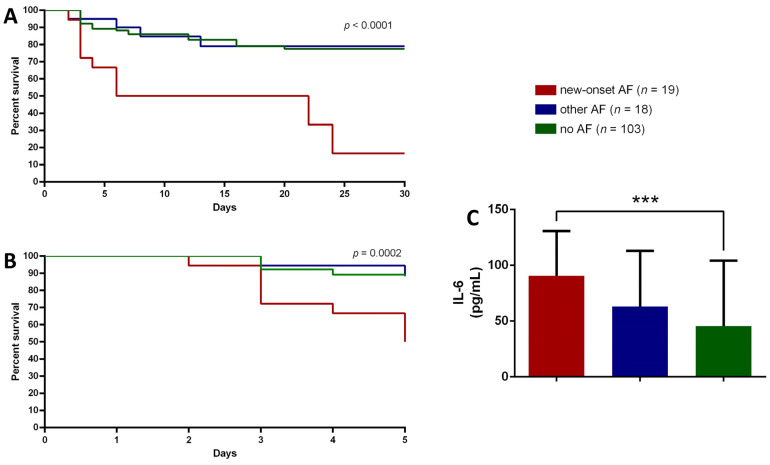
(**A**–**C**). Kaplan-Meier survival plot analysis for hospitalized COVID-19 patients, after the exclusion of participants with PaO^2^/FiO^2^ > 300, with new-onset atrial fibrillation (new-onset AF, *n* = 19, red line), with other forms of AF (other AF, *n* = 18, blue line) and without AF (no AF, *n* = 103, green line) at 28 days (**A**) and at 5 days (**B**). Interleukin 6 (IL-6) serum levels comparison between study groups (**C**), after the exclusion of participants with PaO^2^/FiO^2^ > 300. Data are expressed as median ± interquartile range. *** = *p* < 0.0005.

**Figure 4 medicina-58-00530-f004:**
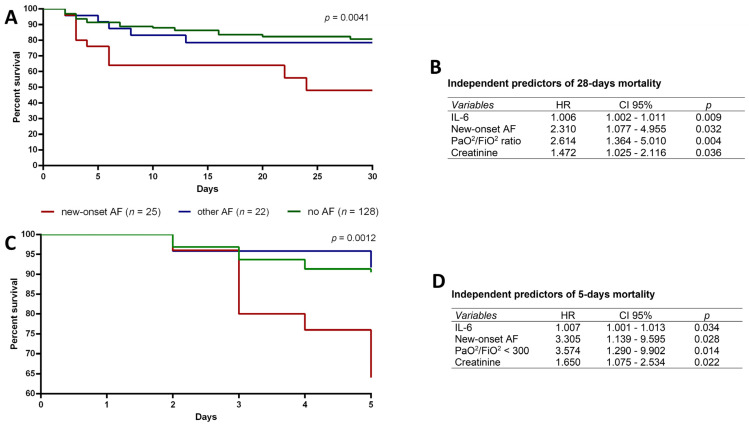
(**A**–**D**). Kaplan-Meier survival plot analysis for hospitalized COVID-19 patients with new-onset atrial fibrillation (new-onset AF, *n* = 25, red line), with other forms of AF (other AF, *n* = 22, blue line) and without AF (no AF, *n* = 128, green line) at 28 days (**A**) and at 5 days (**C**). Cox regression analysis results show that IL-6, creatinine, new-onset atrial fibrillation and PaO^2^/FiO^2^ < 300 are independent predictors of 28-days (**B**) and 5 days mortality (**D**).

**Table 1 medicina-58-00530-t001:** Demographic and clinical features of study population at admission.

	New Onset AF (*n* = 25)	Other AF (*n* = 22)	No AF (*n* = 128)	*p*
*Demographics*				
Age, median (IQR)	81 (73.6–84)	80.5 (70.8–84)	71.5 (57.5–83.9)	**<0.005**
BMI, median (IQR)	25.4 (22.8–27.7)	24.1 (21.7–30.6)	25.1 (22.5–27)	0.712
Smoking, current, *n* (%)	3 (12)	2 (9)	21 (16.4)	0.138
Smoking, past, *n* (%)	2 (8)	3 (13.6)	10 (7.8)	0.744
Gender, male, *n* (%)	12 (48)	12 (54.5)	67 (52.3)	0.571
Hospital stay (days), median, (IQR)	12 (5–27)	22 (13–40.5)	21 (13–36)	0.111
*Comorbidities*				
Diabetes, *n* (%)	5 (20)	4 (18.1)	33 (25.7)	0.629
Coronary artery disease, *n* (%)	6 (24)	6 (27.2)	21 (16.4)	0.228
COPD, *n* (%)	3 (12)	4 (18.1)	12 (9.3)	0.463
Heart failure, *n* (%)	5 (20)	9 (40.9)	17 (13.2)	**<0.005**
Hypertension, *n* (%)	16 (64)	15 (68.1)	78 (60.9)	0.139
Chronic kidney disease, *n* (%)	11 (44)	11 (50)	22 (17.1)	**<0.001**
Cerebrovascular disease, *n* (%)	4 (16)	5 (22.7)	29 (22.6)	0.449
Respiratory failure, *n* (%)	1 (4)	0 (0)	4 (3)	0.575
*Vital and clinical parameters*				
Heart rate, median (IQR)	80 (77.5–95)	80 (72.5–95.5)	80 (76–90)	0.068
DBP mmHg, median (IQR)	70 (60–80)	70 (66-82)	70 (60–78)	0,355
SBP mmHg, median (IQR)	130 (110–150)	125 (120–140)	130 (115–145)	0.541
MBP mmHg, median (IQR)	90 (76–103)	88 (84–99)	91 (80–102)	0.402
Respiratory rate, median (IQR)	23 (18–28)	19 (18–25)	18 (17–22)	**<0.001**
Fever, *n* (%)	4 (16)	6 (22.7)	56 (43)	**<0.05**
PaO^2^/FiO^2^ ratio, median (IQR)	286 (147–366)	320 (197–388)	309 (214–357)	0.665
CHA_2_DS_2_-VASc, median (IQR)	3 (2–4)	3 (2–4)	-	0.4
HAS-BLED, median (IQR)	1 (1–2)	1 (1–2)	-	0.94
*Therapy*				
Corticosteroid, *n* (%)	23 (92)	19 (86)	109 (85)	0.66
COT, *n* (%)	13	10	73	0.59
HFNO, *n* (%)	5	5	26	0.96
c-PAP, *n* (%)	6	3	24	0.67
NIMV, *n* (%)	1	1	5	0.97

Data are expressed as median ± interquartile range (IQR) or number with percentage accordingly; SBP: systolic blood pressure; DBP: diastolic blood pressure; MBP: medium blood pressure; BMI: body mass index; COPD: chronic obstructive pulmonary disease; CHA_2_DS_2_-VASc: score for stroke risk; HAS-BLED: score for major *bleeding* risk; COT: conventional oxygen therapy; HFNO: high-flow nasal oxygen; c-PAP: continuous positive airway pressure; NIMV: non-invasive mechanic ventilation.

**Table 2 medicina-58-00530-t002:** Laboratory profile of study population at admission.

	New-Onset AF (*n* = 25)	Other AF (*n* = 22)	No AF (*n* = 128)	*p*
*Laboratory findings*				
Albumin, *g/dL*	2.93 (2.7–3.23)	3 (2.72–3.35)	3.17 (2.8–3.55)	0.18
ALT, *UI/L*	33 (13–38)	26 (13–25.5)	18.5 (13–34)	0.508
AST, *UI/L*	30 (27–50)	28 (19.25–35.25)	24 (17–36.5)	0.660
CK, *U/L*	63 (38–126)	76 (37–324.5)	103 (46–243.5)	0.369
Creatinine, *mg/dL*	1.5 (0.9–2.3)	0.9 (0.7–1.5)	0.9 (0.7–1.2)	**0.005**
D-DIMER, *mcg/mL*	1.44 (0.48–4)	1 (0.6–1-6)	1 (0.485–1.88)	0.32
Fibrinogen, *mg/dL*	503 (320–660)	476.5 (345–564)	527 (418–637)	0.16
Hb, *gr/dL*	12.4 (10.7–13.7)	12.2 (10.5–13.9)	13 (11–14.5)	0.142
IL-6, *pg/mL*	80.9 (54–130)	27.5 (12.9–40.4)	20.3 (8.4–38.8)	**0.001**
LDH, *U/L*	654 (331–516)	422,5 (476–806)	383 (323–533)	0.478
Lymphocyte count, *cells*	816 (600–1503)	1308 (1143–1576)	1376.5 (842–1815.5)	0.173
NT-PRO-BNP, *pg/mL*	1939 (488.75–2946)	1972 (1402–4907)	196 (75–996)	0.197
PCR, *mg/dL*	6.9 (0.96–13.6)	2.95 (1.18–6.55)	3.5 (0.7–7.9)	0.712
PCT, *ng/mL*	0.2 (0.09–0.31)	0.09 (0.04–0.16)	0.09 (0.05–0.21)	0.654
PLT, *cells × 10^4^*	155 (90–172)	166 (136–242)	206 (155–252)	**0.017**
Troponin, *pg/mL*	121 (98–259)	200 (41–536)	34.57 (11–137)	0.616
Urea, *mg/dL*	63 (42–108)	38 (27.25–61.75)	40.5 (28.25–70)	**0.01**
WBC, *cells*	9700 (4400–14940)	6000 (4725–11325)	6850 (4900–9975)	0.365

Data are expressed as the median ± interquartile range (IQR); WBC: white blood cells; PLT: platelet cells; PCT: procalcitonin; Hb: haemoglobin; CRP: C-reactive protein; IL-6: interleukin 6; NT-PRO-BNP: N-terminal pro-brain natriuretic peptide; LDH: lactate dehydrogenase; CK: creatine kinase; AST: aspartate transaminase; ALT: alanine transaminase.

## Data Availability

The datasets used and/or analysed during the current study are available from the corresponding author on reasonable request.

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
