# Peer review of "New-Onset Atrial Fibrillation and Early Mortality Rate in COVID-19 Patients: Association with IL-6 Serum Levels and Respiratory Distress"

_medicina, 2022, doi:10.3390/medicina58040530_

Round 1

Reviewer 1 Report

Bagnato et.al. presented an interesting study including hospitalized COVID-19 patients with evidence of pneumonia on CT and found that new-onset atrial fibrillation has been associated with higher IL-6 levels and short-term mortality. The study design was very clear and straightforward. And authors have presented their results in an organized format. However, I have a few suggestions.

  1. It would be better to show the cause of mortality was whether cardiovascular or non-cardiovascular.
  2. New-onset atrial fibrillation group was noted to have elevated levels of creatinine and Urea, suggesting a higher incidence of acute kidney injury. Prior literature found that acute kidney injury is associated with higher mortality for COVID-19 patients. Adding creatinine into the Cox regression model to investigate predictors of mortality may potentially reduce confounding bias.
  3. Table 1 presented information regarding "Vitals and clinical parameters". Were those values on admission or the average of hospitalization?
  4. Treatment information was missing in the manuscript. I would recommend authors present data regarding steroid use, mechanical ventilation if possible.

Author Response

We would first like to thank the reviewer for the valuable comments to our manuscript:

1Q: It would be better to show the cause of mortality was whether cardiovascular or non-cardiovascular.

1A: This is a very good observation. However, we focused our attention on the relevant acute cardiovascular and cerebrovascular events observed at admission. Unfortunately, the precise distinction of the mortality cause is not applicable to the actual study due to the nature of the design.

2Q: New-onset atrial fibrillation group was noted to have elevated levels of creatinine and Urea, suggesting a higher incidence of acute kidney injury. Prior literature found that acute kidney injury is associated with higher mortality for COVID-19 patients. Adding creatinine into the Cox regression model to investigate predictors of mortality may potentially reduce confounding bias.

2A: this is a very interesting suggestion indeed. We performed the Cox regression analysis with creatinine as a covariate and the new results are reported in the text and figure.

3Q: Table 1 presented information regarding "Vitals and clinical parameters". Were those values on admission or the average of hospitalization?

3A: Vitals and clinical parameters were collected at admission. This has been specified in the table title and also in the results section following your suggestion.

4Q: Treatment information was missing in the manuscript. I would recommend authors present data regarding steroid use, mechanical ventilation if possible.

4A: Treatment information has been added in table and in the text for oxygen therapy and steroid use as requested.

Reviewer 2 Report

This is a study whose objective was to analyze the impact of new-onset Interleukin-6 (IL-6) and Atrial Fibrillation (AF) on the mortality rate in hospitalized patients with COVID-19 stratified by PaO2/FiO2. These findings suggest that the increase in hospital mortality depends on the severity of the disease and worsens in the presence of new-onset AF and rising serum IL-6 levels.

The originality of the work lies in having identified that high serum levels of IL-6 is a poor prognostic factor in patients admitted for COVID-19 who present with new-onset AF.

The main limitations of the work have been commented by the authors in the work, but another limitation of the same, not negligible, is the low number of patients, although this has not made it difficult to find significant differences between the risk groups and the mortality to 5 and 28 days.

The authors adequately discuss these findings and the possible role of IL-6 with neurological events in patients with new-onset AF and acknowledge that this needs to be confirmed. They also comment that the association between a proinflammatory state and the appearance of AF in viral diseases could be attributed to a greater release of cytokines, mainly IL-6.

The conclusions are a bit long, we suggest moving the following paragraph to the discussion and removing it from the conclusions: “Palpitations are one of the main warning symptoms and require medical personnel to immediately suspect COVID-19, even in the absence of ARDS or the most common presenting symptoms of COVID.” In addition, this conclusion was not contemplated in the proposed objectives.

Author Response

We would first like to thank the reviewer for the valuable comments to our manuscript. The phrase in the conclusion was removed as a similar sentence is already present in the introduction. We confirm that the limitations highlighted are reported in the discussion as requested.